# Isolation and Characterization of Cross-Reactive Human Monoclonal Antibodies That Potently Neutralize Australian Bat Lyssavirus Variants and Other Phylogroup 1 Lyssaviruses

**DOI:** 10.3390/v13030391

**Published:** 2021-03-01

**Authors:** Dawn L. Weir, Si’Ana A. Coggins, Bang K. Vu, Jessica Coertse, Lianying Yan, Ina L. Smith, Eric D. Laing, Wanda Markotter, Christopher C. Broder, Brian C. Schaefer

**Affiliations:** 1Department of Microbiology, Uniformed Services University, Bethesda, MD 20814, USA; dawn.l.weir.mil@mail.mil (D.L.W.); siana.coggins.ctr@usuhs.edu (S.A.C.); Bangvu@ymail.com (B.K.V.); lianying.yan.ctr@usuhs.edu (L.Y.); eric.laing@usuhs.edu (E.D.L.); 2Henry M. Jackson Foundation for the Advancement of Military Medicine, Bethesda, MD 20814, USA; 3Department of Medical Virology, Faculty of Health Sciences, University of Pretoria, Pretoria, Gauteng 0001, South Africa; jessicac@nicd.ac.za (J.C.); wanda.markotter@up.ac.za (W.M.); 4Centre for Emerging Zoonotic and Parasitic Diseases, National Institute for Communicable Diseases, National Health Laboratory Service, Sandringham, Johannesburg 2131, South Africa; 5Risk Evaluation and Preparedness Program, Health and Biosecurity, CSIRO, Black Mountain, Australian Capital Territory 2601, Canberra, ACT 2601, Australia; Ina.smith@csiro.au

**Keywords:** bat, monoclonal antibodies, lyssaviruses, neutralization, glycoprotein, ABLV, rabies, RABV, phage display

## Abstract

Australian bat lyssavirus (ABLV) is a rhabdovirus that circulates in four species of pteropid bats (ABLVp) and the yellow-bellied sheath-tailed bat (ABLVs) in mainland Australia. In the three confirmed human cases of ABLV, rabies illness preceded fatality. As with rabies virus (RABV), post-exposure prophylaxis (PEP) for potential ABLV infections consists of wound cleansing, administration of the rabies vaccine and injection of rabies immunoglobulin (RIG) proximal to the wound. Despite the efficacy of PEP, the inaccessibility of human RIG (HRIG) in the developing world and the high immunogenicity of equine RIG (ERIG) has led to consideration of human monoclonal antibodies (hmAbs) as a passive immunization option that offers enhanced safety and specificity. Using a recombinant vesicular stomatitis virus (rVSV) expressing the glycoprotein (G) protein of ABLVs and phage display, we identified two hmAbs, A6 and F11, which completely neutralize ABLVs/ABLVp, and RABV at concentrations ranging from 0.39 and 6.25 µg/mL and 0.19 and 0.39 µg/mL respectively. A6 and F11 recognize overlapping epitopes in the lyssavirus G protein, effectively neutralizing phylogroup 1 lyssaviruses, while having little effect on phylogroup 2 and non-grouped diverse lyssaviruses. These results suggest that A6 and F11 could be effective therapeutic and diagnostic tools for phylogroup 1 lyssavirus infections.

## 1. Introduction

Australian bat lyssavirus (ABLV) was first isolated in 1996 from a grounded black flying fox (*Pteropus alecto*) found near Ballina, Australia [1]. Since then, ABLV has been isolated from all four mainland species of flying foxes (*Pteropodidae* family) as well as the yellow-bellied sheath-tailed bat (*Saccolaimus flaviventris*), with two genetically distinct lineages circulating in frugivorous (genus *Pteropus*, ABLVp) [2] and insectivorous (genus *Saccolaimus*, ABLVs) [3] Australian bat populations. Before the discovery of ABLV, Australia was thought to be devoid of endemic lyssaviruses. Biosurveillance projects over the years have drastically expanded the number of known ABLV isolates and provided serological evidence of ABLV infection in a variety of Australian microbat populations [4]. While the prevalence of ABLV antigen, indicative of active infection, is <1% in wild bat populations, increased prevalence is observed in wounded, sick, and orphaned bats [4,5]. Indeed, a recent study found that flying fox pups are a uniquely vulnerable group that is potentially at an heightened risk for mass infection [6]. ABLV can be transmitted to humans from a scratch or bite originating from an infected animal. Historically, there have been three documented human ABLV cases [7,8,9,10], all of which manifested as fatal acute encephalitis that presented after variable periods of incubation following the exposure event (5 weeks to 2 years) (reviewed in [11]). In addition to the documented human infections, ABLV was also isolated from two fatal horse infections in Australia in 2013 [12]. 

Taxonomically, ABLV is a rhabdovirus that belongs to the genus *Lyssavirus*, a group of 17 viral species with the majority having ancestral origins in bats (order *Chiroptera*). All lyssavirus species are capable of causing fatal neurological disease with symptomatic presentation and disease progression that is indistinguishable from clinical rabies. Phylogenetic analyses have enabled the subdivision of lyssavirus isolates into at least two phylogroups and several ungrouped viruses [13,14]. Phylogroup I includes the prototype lyssavirus, *Rabies lyssavirus* (RABV), ABLV, *Duvenhage lyssavirus* (DUVV), *Aravan lyssavirus* (ARAV), *Bokeloh bat lyssavirus* (BBLV), *Irkut lyssavirus* (IRKV), *Khujand lyssavirus* (KHUV), *Gannoruwa bat lyssavirus* (GBLV), *European bat 1 lyssavirus* (EBLV-1), *European bat 2 lyssavirus* (EBLV-2), *Taiwan bat lyssavirus* (TWBLV), and *Kotalahti bat virus* (KBLV). *Shimoni bat lyssavirus* (SHIBV), *Lagos bat lyssavirus* (LBV), and *Mokola lyssavirus* form phylogroup II. Finally, the most genetically divergent lyssaviruses are ungrouped and include *West Caucasian bat lyssavirus* (WCBV), *Ikoma lyssavirus* (IKOV), and *Lleida bat lyssavirus* (LLEBV) [14]. While genetically and serologically distinct from one another, all lyssaviruses are enveloped bullet-shaped viruses with 12 kb negative-sense single-stranded RNA genomes that encode five major viral proteins: nucleoprotein (N), phosphoprotein (P), matrix (M), glycoprotein (G), and viral RNA polymerase (L) [15]. Lyssavirus G monomers organize in trimers on the virion surface, mediating viral attachment to host cell receptors and facilitating the subsequent clathrin-dependent fusion of viral and host cell membranes during viral entry [16,17,18]. As a surface-expressed viral protein, G is typically the sole target of neutralizing antibodies against lyssaviruses [19,20]. Despite this fact, cross-neutralization between lyssavirus phylogroups is limited, likely due to the high genetic diversity of lyssavirus G sequences [13,21,22,23].

Following any lyssavirus exposure event, prompt administration of the RABV post-exposure prophylaxis (PEP) protocol is highly recommended. PEP consists of thorough cleansing of the wound area followed by administration of the rabies vaccine and rabies immunoglobulin (RIG) (reviewed in [24]). Currently, there are two species of RIG available for post-exposure management: human RIG (HRIG) and equine RIG (ERIG). While HRIG is safe and effective when included in PEP, supply limitations and high production costs have made this resource widely inaccessible. ERIG is occasionally used to replace HRIG in PEP, however the high immunogenicity of this therapeutic is the cause of substantial safety concerns [25,26], with documented cases of ERIG-associated serum sickness [27]. The absence of a safe, well-sourced passive immunization component in PEP has led many to propose the replacement of RIG with virus-neutralizing human monoclonal antibodies (hmAbs) [28,29]. Here, we developed hmAbs specific for ABLV by using a recombinant vesicular stomatitis virus (rVSV) in which VSV G was replaced by G from ABLVs. This virus was employed as the capture antigen for panning of a naïve human antibody fragment (Fab) library. This screen resulted in identification of two antigen binding fragments (Fabs), F11 and A6, with specific binding to ABLV G. These Fabs were further engineered to generate human IgG1 monoclonal antibodies (hmAbs). We report that A6 and F11 are cross-reactive hmAbs that potently neutralize both ABLV variants, RABV, and other phylogroup I lyssaviruses. 

## 2. Materials and Methods

### 2.1. Cells and Viruses

HEK293T cells were provided by Gerald Quinnan (Uniformed Services University) and were maintained in Dulbecco’s modified Eagle’s medium (DMEM; Quality Biologicals, Gaithersburg, MD) supplemented with 10% cosmic calf serum (CCS) (Hyclone, Logan, UT) and 2 mM L-glutamine (DMEM-10). Recombinant turbo green fluorescent protein (GFP) expressing vesicular stomatitis viruses (rVSV) that express ABLVs G, ABLVp G, Rabies CVS-11 G, and VSV (Indiana) G glycoproteins, and rABLVp-GFP have been previously described [30,31]. A global representative panel of lyssaviruses representing all phylogroups was included in virus neutralization testing.

### 2.2. Phage Panning

A previously prepared naïve human Fab phage display library (a total diversity of about 10^10^ members) was used for selection of Fabs; a gift from Dr. Dimiter S. Dimitrov, University of Pittsburgh Medical School) [32,33]. One milliliter of the library was re-amplified in *E. coli* TG1 cells and panning with 1 × 10^12^ phage was carried out as previously described [33]. In brief, antigen (10^6^ plaque forming units (PFU) of a recombinant vesicular stomatitis virus (rVSV) encoding the ABLVs G gene from an isolate of ABLV derived from a yellow-bellied sheath-tailed bat (VSV-ABLVs-G) [3] was coated in 100 µl with PBS pH 7.4 on a high-adsorbing flat bottom 96-well plate and incubated overnight at 4 °C. Recovered phage was mixed with TG1 cells for 1 hour at 37 °C, and the phage was amplified from the infected cells and used in the next round of biopanning. After 3 rounds of biopanning, the recovered enriched phage was evaluated by ELISA; in brief, antigen (5 × 10^4^ PFU VSV-ABLVs-G) was coated on ELISA plates, and recovered phage from the 1st, the 2nd and the 3rd rounds were checked at 2 × 10^10^/well. Following binding and washing, phage were detected by using horseradish peroxidase (HRP)-conjugated goat anti-M13 antibody (GE Healthcare, USA, Chicago, IL, USA). The plates were then washed again to remove non-specifically bound antibody, the ABTS [2,2′-azinobis (3-ethylbenzothiazoline-6-sulfonic acid)] substrate (Roche, Indianapolis, IN, USA) was added, and the solution absorbance at 405 nm (A405) was measured.

After confirming phage panning enrichment specific to the VSV-ABLVs-G antigen at the 3^rd^ round of panning, about 40–50 clones were randomly picked from infected TG1 colonies on agar plates for monoclonal phage ELISA; in brief, single colonies were grown in 96-well plates in 150 µL 2YT medium with 0.02% glucose and ampicillin for 2 hours with shaking at 37 °C (a double plate was made for back up clones and kept aside at 4 °C); 25 µL of helper phage M13K07 in 2YT medium (total 10^9^) was added into each well, and the plate was incubated for 30 min at 30 °C. Then, 25 µL of 2YT medium containing ampicillin 100 µg/mL and kanamycin 200 µg/mL was added into each well and further incubated overnight at 30 °C in a shaker at 200 rpm for phage secretion. The next day, the plate was centrifuged at 4000 rpm for 12 min. The phage supernatants were recovered and assayed by phage ELISA as above. The positive (A405 > 0.8 OD) clones from the back-up plate were grown for extraction of phagemid DNA and sequencing. Upon analysis of sequencing results of the monophage ELISA, several positive clones were identical, and 2 Fab clones (A6 and F11) were obtained.

### 2.3. Isolation and Characterization of hmAbs F11 and A6

The variable regions (V_H_ and V_L_) of positive clones were sequenced and used to express and purify Fabs. The V_H_ and V_L_ gene segments were then cloned into the human IgG1 expression vector pDR12 (provided by D. Burton, Scripps Research Institute, La Jolla, CA, USA), yielding the constructs that were used to produce the IgG1 hmAbs, F11 and A6. Stable expression in 293F cells was achieved by re-cloning the IgG1 construct into pcDNA3.1 Hygro-B and selection of cell lines with Hygromycin at 200 µg/mL. IgG1 hmAbs A6 and F11 were purified by Protein G Sepharose affinity chromatography from culture supernatants.

To analyze the antigen-binding activity of the purified hmAbs, Immulon 2HB microtiter ELISA plates (Fisher Scientific, Hampton, NH, USA) were coated overnight at 4 °C with 10^4^ fluorescence focus units (FFU) [34] of rVSV-ABLVs-G, rVSV-ABLVp-G, or rVSV-VSV-G per well diluted in 1×PBS. Plates were blocked with 1×PBS containing 5% bovine serum albumin (BSA) and 0.05% Tween-20 (BSA-PBST) for 1 hr at 37 °C. Human mAbs were diluted in 1% BSA-PBST in 2-fold series and were assayed in duplicate. Goat anti-human IgG HRP (Thermo Fisher Scientific, Waltham, MA, USA) was used for detection. For each step, plates were incubated at 37 °C for 1 h and subsequently washed 6 times with PBST. Plates were incubated with ABTS substrate (100 µL per well) for 30 min with shaking at room temperature. The absorbance was measured for each well at 405 nm, and the average value was calculated from duplicates. 

### 2.4. Virus Neutralization Assays

Purified hmAbs were serially diluted in DMEM-10, in duplicate wells, in a 96-well tissue culture plate and mixed with 5 × 10^4^ FFU of either VSV-ABLVs-G-GFP, VSV-ABLVp-G-GFP, VSV-RABV-G-GFP, VSV-VSV-G-GFP, or ABLVp-GFP reporter viruses for 30 min at 37 °C. Dilutions of purified hmAbs started at 25 µg/mL or 10 µg/mL, respectively, for recombinant VSV reporter viruses and ABLVp-GFP virus. A total of 5 × 10^4^ HEK293T cells were added to each well, followed by incubation at 37 °C for 20 h (VSV recombinant reporter viruses) or 48 h (ABLVp-GFP). Wells were then scored for GFP expression. Neutralization titers were recorded as the hmAb concentration where at least one of the duplicate wells was positive for GFP expression. Single-cell preparations were made and fixed (2% paraformaldehyde in 1×PBS) and GFP expression, indicative of productive infection, was analyzed by a Nexcelom Vision Cellometer (Nexcelom Bioscience LLC., Lawrence, MA, USA) capable of fluorescence detection. The percent of infected cells was calculated by dividing the number of GFP positive cells by the total number of cells and multiplying by 100. Results are expressed as percent inhibition relative to that of untreated controls. Error bars represent the standard error of the mean (SEM). Greater than 90% of untreated cells were infected after 20 h.

### 2.5. Competitive ELISA

Biotinylated hmAbs A6 and F11 were prepared using NHS-PEO-biotin bound to a nickel chelated support matrix according to the manufacturer’s directions (Pierce, Rockford, IL, USA). Immulon 2HB microtiter ELISA plates (Fisher Scientific, Hampton, NH, USA) were coated overnight at 4 °C with 10^4^ FFU rVSV-ABLVs-G per well, diluted in PBS. Plates were blocked with PBS containing 5% bovine serum albumin (BSA) and 0.05% Tween-20 (BSA-PBST) for 1 hr at 37 °C. Unlabeled hmAbs were diluted in 1% BSA-PBST in 2-fold series starting at 16 µg/mL and were assayed in duplicate. Plates were incubated at room temperature for 30 min and, subsequently, 25 µL of biotinylated hmAb (1 µg/mL) was added to wells and incubated at 37 °C for 30 min. Following incubation, 50 µL of HRP conjugated streptavidin (Pierce, Rockford, IL, USA) was added at a final dilution of 1:5000 in 1% BSA-PBST and plates were incubated for 1 h at 37 °C. For each step, plates were washed 6 times with PBST. Plates were incubated with ABTS substrate (100 µL per well) for 30 min with shaking at room temperature. The absorbance was measured for each well at 405 nm, and the average value was calculated from duplicates.

### 2.6. Lyssavirus Neutralizations

The in vitro neutralization activity of hmAb A6 and F11 were determined using a modified micro-neutralization test [35,36]. The test was performed in a humidity chamber on 8-well cell culture slides (Marienfeld, Germany). Briefly, 1.75 µL of A6 (0.71 mg/mL) and F11 (0.469 mg/mL) were 5-fold serially diluted in 7 µL of Dulbecco’s Modified Eagle Medium with Ham’s F12 (DMEM/F12, Gibco, Life Technologies, Carlsbad, CA, USA) supplemented with 10% fetal bovine serum (FBS, Gibco, Life Technologies, Carlsbad, CA, USA) and 1% antibiotics (100 U/mL Penicillin, 100 µg/mL Streptomycin and 0.25 µg/mL Amphotericin, Lonza, Switzerland) in each well of the 8-well slides. Wells represented dilutions from 1:10 through 1:78, 1250. To each well, 7 µL of challenge virus 50 focus-forming dose (50FFD_50_), as determined by titration [37] was added and incubated at 37 °C with 5% CO_2_ for 90 min. Back titration of the challenge viruses and cell-only control were completed on a separate 8-well cell culture slide. After incubation, 14 µL of mouse neuroblastoma cells (C1300 clone) was added to each well. Slides were incubated at 37 °C with 5% CO_2_ for 20 h. Slides were fixed with cold acetone for 30 min and stained with 1:100 diluted FITC-anti-lyssavirus conjugate (Agricultural Research Council-Onderstepoort Veterinary Institute, South Africa) with 0.2% Evans blue as counterstain. In each well, 10 fields at 200× magnification were scored based on the presence/absence of fluorescent foci. All tests were performed in triplicate. The dilutions reported represent the highest dilution where less than 50% of the observed fields contained infected cells (i.e., the 50% end-point titer or IC_50_).

### 2.7. Statistics

One-way ANOVA analysis with Dunnett’s multiple comparisons was performed to evaluate hmAb A6 and F11 binding differences between ABLVp G and ABLVs G, and VSV G. Two-way ANOVA analysis with Tukey’s multiplex comparisons was performed to identify significant differences between A6/F11 and m102.4. Significant inhibitory activity of A6 and F11 against rVSV-GFP-G variants was determined via two-way ANOVA and Sidak’s multiple comparisons test of hmAb concentration to 0 µg/mL. Figures and statistical analysis were generated using GraphPad Prism version 7.0. 

## 3. Results

### 3.1. Identification of Phage-Displayed Fabs A6 and F11 That Are Specific for ABLVs Glycoprotein 

We previously reported the isolation of potent henipavirus-neutralizing hmAb m102 through the screening of a large naïve human Fab library against the soluble HeV G glycoprotein [32,33]. Here, we used the same phage library, which contains over 10^10^ phage-displayed human Fabs, to identify Fabs that are specific for ABLVs glycoprotein (ABLVs G). Since soluble ABLV G remains unavailable, we used a turbo green fluorescent protein (GFP)-encoding replication competent recombinant vesicular stomatitis virus (rVSV-∆G) that expresses ABLVs G (rVSV-ABLVs-G) as the antigen for screening the Fab library. Three rounds of phage panning against rVSV-ABLVs-G recombinant reporter virus resulted in the enrichment of anti-ABLVs-G Fabs A6 and F11. To probe the binding activity of the identified Fabs, we performed phage ELISAs using rVSV-ABLVs-G coated wells (see Section 2). 

Fabs A6 and F11 displayed significant binding to ABLVs G, with A6 exhibiting stronger binding to ABLVs G than F11 (Figure 1A). Sequencing revealed both anti-ABLVs G Fabs have unique amino acid sequences that differ by at least one residue in 5 of the 6 Fab complementarity determining regions (CDRs). Notably, the light chain (LC) CDRs exhibited more variability than the heavy chain (HC) CDRs, with CDR-H1 being identical between F11 and A6, and CDR-H2 and CDR-H3 differing by two and one residues, respectively (Figure 1B). 

The DNA sequences encoding the LC and HC of Fabs A6 and F11 were then cloned into the CMVp-driven expression vector, pDR12, for conversion to a whole antibody format and expression as IgG1 hmAbs. To test the antigen-binding activity of the purified hmAbs, we conducted ELISAs using recombinant VSV-∆G viruses expressing G from ABLVs, ABLVp, or VSV as the well-coating antigen. 

We found that purified A6 binds ABLVs G and ABLVp G with similar strength while displaying negligible binding to VSV G (Figure 2A). Interestingly, unlike A6, slight variation was observed in the maximal binding of F11 to ABLVs and ABLVp G variants (Figure 2B). Overall, however, these data show that both hmAbs A6 and F11 exhibit strong binding to ABLVs G and ABLVp G and minimal binding to VSV G. 

Competition ELISAs using rVSV-ABLVs-G coated plates and biotinylated hmAbs revealed that A6 (Figure 3A) and F11 (Figure 3B) recognize overlapping epitopes on ABLV G. These results demonstrated the successful conversion of anti-ABLVs G Fabs A6 and F11 into functional IgG1 hmAbs that recognize overlapping epitopes on both ABLVs G and ABLVp G. 

### 3.2. Neutralization of Recombinant ABLV Variants and RABV by hmAbs A6 and F11

To test the neutralization activity hmAbs A6 and F11, we first measured their ability to inhibit the infection of HEK293T cells by an ABLVp-GFP reporter virus (rABLVp-GFP), using a virus neutralization assay (see Section 2).

Results from duplicate experiments showed that A6 and F11 inhibited viral infection (as evidenced by prevention of GFP expression) when added to cultures at final concentrations of at 0.31 µg/mL and 0.16 µg/mL respectively (Table 1). m102.4, an antibody known to neutralize Hendra virus (HeV) and Nipah virus (NiV) through binding of surface glycoprotein G [32], did not neutralize rABLVp-GFP infection. While phage display was performed using rVSV-ABLVs-G recombinant virus for Fab selection, these data show that hmAbs A6 and F11 possess the ability to neutralize ABLVp infections in vitro.

Since ABLV is the closest genetic relative to RABV, we next tested the ability of the hmAbs to inhibit the infection of HEK293Tcells, using rVSV-∆G reporter viruses expressing ABLVs, ABLVp, or RABV G (Figure 4). 

Recombinant GFP virus expressing ABLVs G (rVSV-ABLVs-G) displayed less than 5% inhibition of infection in the presence of 0.10 µg/mL of either hmAb. Indeed, greater than 50% inhibition of ABLVs-recombinant virus required 0.19 µg/mL A6 and 0.39 µg/mL F11 (Figure 4A). Conversely, greater than 50% inhibition of rVSV-ABLVp-G (Figure 4B) and rVSV-RABV-G (Figure 4C) infections was achieved by treatment with 0.05 µg/mL A6 or F11.

Furthermore, as detailed in Table 2, A6 neutralized 100% of recombinant rVSV-∆G reporter viruses expressing ABLVs G (3.12 µg/mL), ABLVp G (0.39 µg/mL), and RABV G (0.19 µg/mL) at lower concentrations than those observed for F11 (6.25, 0.39, and 0.39 µg/mL respectively). Taken together, these results demonstrate that hmAbs A6 and F11 are potent cross-reactive antibodies that can neutralize both known ABLV variants as well as RABV CVS variant, another phylogroup I virus. 

To more broadly characterize the neutralization activity of A6 and F11, we evaluated the neutralization of a panel of diverse lyssaviruses, including members of phylogroup II (Table 3). 

Consistent with the above data, ABLV and several distinct RABV isolates were potently neutralized by A6 and F11. Similarly, phylogroup I lyssaviruses EBLV-1, EBLV-2, ARAV, DUVV, KHUV, and IRKV were also neutralized by the anti-ABLV-G hmAbs. In fact, A6 and F11 neutralized all tested phylogroup I lyssaviruses with IC_50_ values of 0.91 ng/mL and 0.60 ng/mL respectively. As expected, lyssaviruses belonging to phylogroup II (MOKV, SHIBV, and three lineages of LBV) and ungrouped lyssaviruses (IKOV and WCBV) were not neutralized by A6 or F11, indicating that A6 and F11 likely only recognize and neutralize phylogroup I lyssaviruses. 

## 4. Discussion

In this study, we identified two anti-ABLVs G hmAbs, A6 and F11, which potently cross-neutralize both ABLV variants as well as other phylogroup I lyssaviruses. While sequencing shows A6 and F11 are genetically distinct, competitive ELISA results suggest they bind overlapping epitopes. Extended viral passaging in the presence of each hmAb, followed by sequencing of the resulting escape mutants, could help map the precise location of the epitopes bound by these cross-reactive hmAbs. Furthermore, the in vivo activity of A6 and F11 against lyssavirus infections can be further elucidated through their use within our established pre-clinical lyssavirus mouse model [34]. 

The use of polyclonal RIG as the passive immunization component of PEP has presented a variety of complications in areas ranging from safety to accessibility. Unlike RIG, which is derived from the pooled serum of rabies-immune human donors or horses, recombinant virus-neutralizing hmAbs are produced in human cells thus eliminating safety issues associated with blood- and animal-derived antibodies. Recombinant hmAb preparations can be produced in affordable large-scale quantities and assayed in vitro to ensure minimal variation. Replacement of RIG with hmAbs in the PEP protocol could potentially alleviate supply limitations and extend the availability of a complete PEP protocol to more individuals worldwide. 

Various combinations of mAbs can be administered concurrently in antibody cocktails—a treatment that, like RIG, mimics the broad polyclonal antibody response observed in natural infections. The distinct viral epitopes recognized by the different mAbs in a single antibody cocktail present the unique ability to tailor viral counteraction measures and protect against the emergence of resistance variants. The steady progression of virus-specific mAb cocktail development is well exemplified by work done on Ebola virus (EBOV) [39,40]. Indeed, REGN-EB3, a three-mAb cocktail against EBOV G was recently approved by the FDA as a treatment for EBOV infection [41]. Similar to EBOV, several mAb cocktails have been developed for use against RABV infections [28,42,43,44,45,46,47]. In phase 2/3 clinical trials, PEP regimens containing anti-rabies mAb cocktail TwinrabTM (docaravimab and miromavimab) [48] or mAb SII RMAb (Rabishield) [49] demonstrated noninferiority to HRIG in the window of protection and rabies virus neutralizing activity respectively. However, since mAb cocktail therapy is still a developing field, it is important to consider potential escape mutants that result from cocktail treatments [50,51]. Ultimately, to achieve the greatest impact, extensive scientific characterization of anti-lyssavirus hmAbs must be met with an international effort to not only produce these reagents in high quantity, but also provide them to the public at minimal cost.

## Figures and Tables

**Figure 1 viruses-13-00391-f001:**
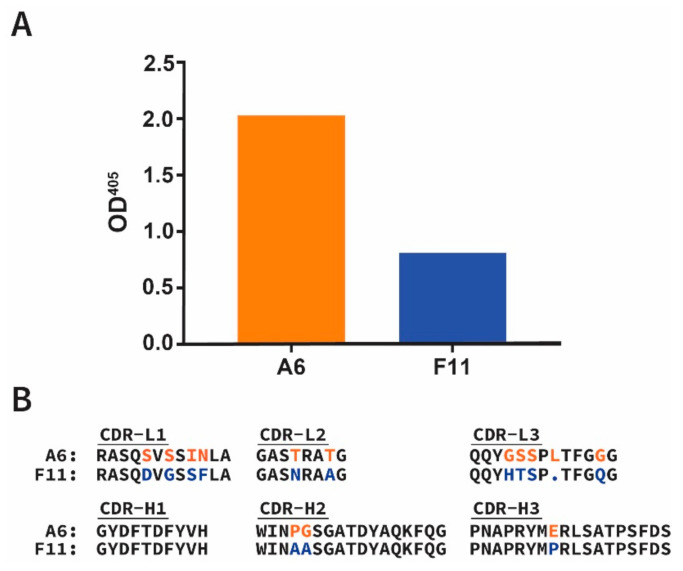
Identification of ABLVs G-binding Fabs A6 and F11. (**A**) Fabs specific for ABLVs G were identified by phage ELISA screening as described in the Section 2. Bound phage were detected by anti-M13 horseradish peroxidase (HRP) conjugated antibody and the resulting solution absorbance at 405 nm shown. (**B**) Amino acid sequences of light chain and heavy chain complementarity determining regions (CDR-L1-3 and CDR-H1-3, respectively) for Fabs A6 and F11. Residues which differ between A6 and F11 CDRs are indicated by red and blue font, respectively.

**Figure 2 viruses-13-00391-f002:**
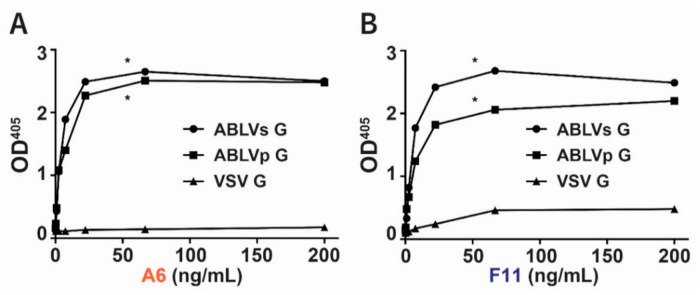
Direct binding of anti-ABLVs G hmAbs to the glycoproteins of ABLVs and ABLVp. The binding of hmAbs (**A**) A6 and (**B**) F11 to ABLVs G (circle), ABLVp G (square), and VSV G (triangle) was quantified using ELISA. The average value from duplicate wells is shown. * *p* < 0.05, one-way ANOVA with Dunnett’s multiple comparisons.

**Figure 3 viruses-13-00391-f003:**
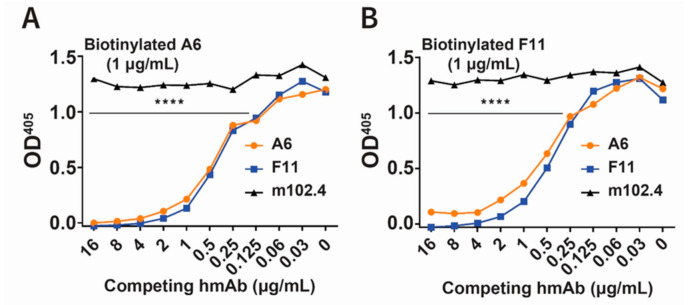
Competitive binding of hmAbs A6 and F11 to ABLVs G. Biotinylated hmAbs (**A**) A6 and (**B**) F11 at 1 μg/mL were used in competition ELISAs employing increasing concentrations of unlabeled hmAbs A6 (orange circle) and F11 (blue square) in rVSV-ABLVs-G coated wells. Anti-henipavirus hmAb m102.4 (black triangle) was included as a negative control. The average value from duplicate wells is shown; **** *p* < 0.0001, two-way ANOVA with Tukey’s multiple comparisons of A6/F11 vs m102.4.

**Figure 4 viruses-13-00391-f004:**
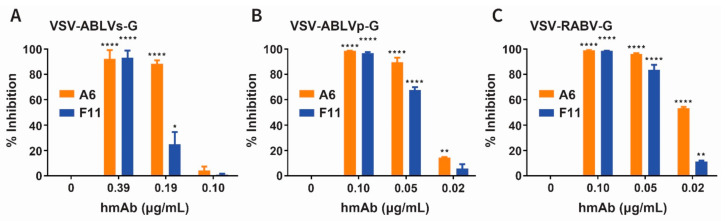
Inhibitory activity of anti-ABLVs G hmAbs A6 and F11 against rVSV-ABLV-G and rVSV-RABV-G infections. HEK239T cells infected with replication-competent recombinant rVSV-GFP viruses expressing (**A**) ABLVs G, (**B**) ABLVp G, and (**C**) RABV G were treated with hmAbs A6 (orange) or F11 (blue). The subsequent percent (%) inhibition was calculated as described in Section 2. Error bars represent the standard error of the mean (SEM); * *p* = 0.05, ** *p* = 0.01, **** *p* < 0.0001, determined by two-way ANOVA with multiple comparisons.

**Table 1 viruses-13-00391-t001:** Neutralization of rABLVp-GFP infection by hmAbs A6 and F11.

	hmAbs^α^							
μg/mL	A6			F11			m102.4	
10	-	-		-	-		+++	+++
5	-	-		-	-		ND	ND
2.5	-	-		-	-		ND	ND
1.25	-	-		-	-		ND	ND
0.62	-	-		-	-		ND	ND
0.31	-	-		-	-		ND	ND
0.16	+	+		-	-		ND	ND
0.08	+	+		+	+		ND	ND
0.04	++	++		++	++		ND	ND
0.02	+++	+++		+++	+++		ND	ND

^α^ Neutralization was performed on HEK293T cells infected with rABLVp-GFP viruses as described in Section 2. (-), no GFP expression; (+) to (+++) indicates the relative intensity of fluorescence; ND, not determined. Results from duplicate experiments are shown.

**Table 2 viruses-13-00391-t002:** Neutralization of rVSV-GFP infection by hmAbs A6 and F11.

[hmAb]^α^μg/mL		ABLVs G						ABLVp G						RABV G						VSV G				
		A6			F11			A6			F11			A6			F11			A6			F11	
25		-	-		-	-		-	-		-	-		-	-		-	-		+++	+++		+++	+++
12.5		-	-		-	-		-	-		-	-		-	-		-	-		ND	ND		ND	ND
6.25		-	-		-	-		-	-		-	-		-	-		-	-		ND	ND		ND	ND
3.12		-	-		+	+		-	-		-	-		-	-		-	-		ND	ND		ND	ND
1.56		+	+		+	+		-	-		-	-		-	-		-	-		ND	ND		ND	ND
0.78		+	+		+	+		-	-		-	-		-	-		-	-		ND	ND		ND	ND
0.39		+	+		+	+		-	-		-	-		-	-		-	-		ND	ND		ND	ND
0.19		+	+		++	++		-	+		+	+		-	-		+	+		ND	ND		ND	ND
0.10		+++	+++		+++	+++		+	+		+	+		+	+		+	+		ND	ND		ND	ND
0.05		+++	+++		+++	+++		+	+		++	++		+	+		+	+		ND	ND		ND	ND
0.02		+++	+++		+++	+++		+++	+++		+++	+++		++	++		+++	+++		ND	ND		ND	ND

^α^ Neutralization was performed on HEK293T cells infected with rVSV-GFP viruses as described in Section 2. (-), no GFP expression; (+) to (+++) indicates the relative intensity of fluorescence; ND, not determined. Results from duplicate experiments are shown.

**Table 3 viruses-13-00391-t003:** Neutralization of diverse lyssaviruses by hmAbs A6 and F11.

			hmAbs ^δ^	
Virus^α^	Accession Number^β^	Phylogroup^γ^	A6	F11
ABLV/*P. alecto*/AUS/1998/RV634	AY062067	I	1:781250	1:781250
RABV/*T. brasiliensis*/USA/2003/FL385	JQ685905	I	1:781250	1:781250
RABV/*D. rotundus*/BRA/1998	AF070449	I	1:781250	1:781250
RABV/*U. cinereoargeteus*/USA/2009/2401	JQ685934	I	1:781250	1:781250
EBLV-1/*E. serotinus*/DNK/1986/RV20	KF155003	I	1:781250	1:781250
EBLV-2 */M. daubentonii*/GBR/1996/RV628	KY688136	I	1:781250	1:781250
ARAV/*M. blythi*/KGZ/1991	EF614259	I	1:781250	1:781250
DUVV/*M. musculus*/RSA/2008/SA06	EU623444	I	1:781250	1:781250
IRKV/*M. leucogaster*/RUS/2003	EF614260	I	1:781250	1:781250
KHUV/*M.* mystacinus/TJK/1992	EF614261	I	1:781250	1:781250
MOKV/*F. catus*/RSA/2014/14/024	KP899612	II	-	-
SHIBV/*H. commersoni*/KEN/2009	GU170201	II	-	-
LBV/*R. aegypticus*/EGY/1999/LBVAfr1999	EF547432	II	-	-
LBV/*E. wahlbergi*/RSA/2016/UP6414	MH643893	II	-	-
LBV/*R. aegypticus*/KEN/2010/KE576	GU170202	II	-	-
IKOV/*C. civetta*/TZA/2009/RV2508	JX193798	U	-	-
WCBV/*M. schreibersi*/2003/RUS	EF614258	U	-	-

^α^ The identity of all isolates was verified using partial N-gene sequencing. Australian bat lyssavirus (ABLV), Rabies lyssavirus (RABV), European bat 1 lyssavirus (EBLV-1), European bat 2 lyssavirus (EBLV-2), Aravan lyssavirus (ARAV), Duvenhage lyssavirus (DUVV), Irkut lyssavirus (IRKV), Khujand lyssavirus (KHUV), Mokola lyssavirus (MOKV), Shimoni lyssavirus (SHIBV), Lagos bat lyssavirus (LBV), Ikoma lyssavirus (IKOV), West Caucasian bat lyssavirus (WCBV). ^β^ Partial N-gene sequences were used to identify the closest match on GenBank. Primers 001lys and 550B [38] were used for sequencing. ^γ^ U, ungrouped. ^δ^ Neutralization was performed on MNA cells infected with diverse lyssaviruses as described in the Section 2. The dilution reported is the 50% end-point titer (IC_50_). (-), no neutralization.

## Data Availability

Not applicable.

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
