# Peer review of "Isolation and Characterization of Cross-Reactive Human Monoclonal Antibodies That Potently Neutralize Australian Bat Lyssavirus Variants and Other Phylogroup 1 Lyssaviruses"

_viruses, 2021, doi:10.3390/v13030391_

Round 1
Reviewer 1 Report
The authors developed a novel approach to characterize the differential immune response in neutralizing antibodies following lyssavirus exposure. The authors are to be congratulated for their timely contribution to the peer-reviewed literature.
Author Response
Reviewer 1
The authors developed a novel approach to characterize the differential immune response in neutralizing antibodies following lyssavirus exposure. The authors are to be congratulated for their timely contribution to the peer-reviewed literature.
We thank the reviewer for the enthusiastic response to this work.
Reviewer 2 Report
This article describes the isolation and characterisation of human monoclonal antibodies against ABLV and cross-neutralisation with other phylogroup I lyssaviruses. The extra new tools for diagnostics and therapeutics are always useful.
I have a few issues with this work, please check:
Line 69, the information is not accurate. Please check ICTV website for the latest information.
In the abstract, higher concentrations of monoclonal antibodies are needed to neutralise RABV than ABLVs/ABLVp; whereas in the result section, it is the other way round.
What is the size of phage Fab library 10 10 or 10 13?
The beginning of section 2.3 seems to be repeating what has been described earlier.
Author Response
Reviewer 2
This article describes the isolation and characterisation of human monoclonal antibodies against ABLV and cross-neutralisation with other phylogroup I lyssaviruses. The extra new tools for diagnostics and therapeutics are always useful.
I have a few issues with this work, please check:
Line 69, the information is not accurate. Please check ICTV website for the latest information.
TWBLV and KBLV have been added to the phylogroup I sentence to reflect the proper classification, as indicated on the ICTV website. Thank you for conveying this updated classification information to us.
In the abstract, higher concentrations of monoclonal antibodies are needed to neutralise RABV than ABLVs/ABLVp; whereas in the result section, it is the other way round.
The abstract has been modified to accurately reflect the data. The sentence in question now reads as follows: “Using a recombinant vesicular stomatitis virus (rVSV) expressing the glycoprotein (G) protein of ABLVs and phage display, we identified two hmAbs, A6 and F11, which completely neutralize ABLVs/ABLVp, and RABV at concentrations ranging from 0.39-6.25 µg/mL and 0.19-0.39 µg/mL respectively.”
What is the size of phage Fab library 10 10 or 10 13?
We thank the reviewer for finding this error. This and additional edits to section 2.2 (methods) have been made. Note, the original diversity of the phage library was 10^10 and the amplified phage library used was 10^12 (not 10^13). Amplifying an aliquot of such an original phage library and making fresh phage is recommended and also increases the number of phage that will be used for panning. In general, the phage used for panning should be at least 100x that of the library size – In this case, the 1ml of phage library used here, with a known library size = 10^10, was amplified and used. Thus, this amplified phage library was then 100 x 10^10 (10^12) for the initial biopanning.
The beginning of section 2.3 seems to be repeating what has been described earlier.
The first three sentences of Section 2.3 have been deleted, resulting in the section beginning with “The variable regions (VH and VL) of positive clones were sequenced...”
Information regarding rVSV-ABLV-G has been moved to the second sentence in Section 2.2. With this edit, the sentence currently reads as follows: “Antigens (106 plaque forming units (PFU) of a recombinant vesicular stomatitis virus (rVSV) encoding the ABLVs G gene from an isolate of ABLV derived from a yellow-bellied sheath-tailed bat [3] (VSV-ABLVs-G)) was coated in 100 µl with PBS pH 7.4 on a high-adsorbing flat bottom 96-well plate - Incubation for overnight at 4 °C.”
Reviewer 3 Report
This is a very well written account of two new potent human monoclonal antibody candidates, proposed for therapeutic use in place of rabies immunoglobulin (RIG), as part of post-exposure prophylaxis and treatment against Australian Bat Lyssavirus and other phylogroup 1 lyssaviruses. The authors consider and reference appropriate background information on the topic to outline the rationale for study. There are a few minor points to address for clarity, but overall nicely done.
L101-108 - please consider adding the details about the various lyssaviruses titrated and evaluated for neutralization activity (i.e., those described in Table 3).
L132 - there is no reference numbered 88 in the literature cited. Can the authors supply the proper reference?
L149 - can the manufacturer of the goat anti-human IgG product be identified if this is a commercially available product?
Author Response
Reviewer 3
This is a very well written account of two new potent human monoclonal antibody candidates, proposed for therapeutic use in place of rabies immunoglobulin (RIG), as part of post-exposure prophylaxis and treatment against Australian Bat Lyssavirus and other phylogroup 1 lyssaviruses. The authors consider and reference appropriate background information on the topic to outline the rationale for study. There are a few minor points to address for clarity, but overall nicely done.
L101-108 - please consider adding the details about the various lyssaviruses titrated and evaluated for neutralization activity (i.e., those described in Table 3).
To accompany the GenBank accession numbers provided for the various lyssaviruses in Table 3, a sentence regarding Table 3 has been added to the end of section 2.1. This sentence reads as follows: “A global representative panel of lyssaviruses representing all phylogroups was included in virus neutralization testing (Table 3).”
L132 - there is no reference numbered 88 in the literature cited. Can the authors supply the proper reference?
Thank you for bringing this error to our attention. The sentence containing this reference has been removed due to its repetitive nature as pointed out by Reviewer 2. The intended references (references 32 and 33) are cited in Section 2.2.
L149 - can the manufacturer of the goat anti-human IgG product be identified if this is a commercially available product?
Thermo Fisher Scientific, Waltham, MA, has been added to the text.